# Clinical Significance of Marginal Zinc Deficiency as a Predictor of Covert Hepatic Encephalopathy in Patients with Liver Cirrhosis

**DOI:** 10.3390/ijms26094184

**Published:** 2025-04-28

**Authors:** Takuya Matsuda, Tadashi Namisaki, Akihiko Shibamoto, Shohei Asada, Fumimasa Tomooka, Takahiro Kubo, Aritoshi Koizumi, Misako Tanaka, Satoshi Iwai, Takashi Inoue, Yuki Tsuji, Yukihisa Fujinaga, Norihisa Nishimura, Shinya Sato, Koh Kitagawa, Kosuke Kaji, Akira Mitoro, Kiyoshi Asada, Hiroaki Takaya, Ryuichi Noguchi, Takemi Akahane, Hitoshi Yoshiji

**Affiliations:** 1Department of Gastroenterology, Nara Medical University, 840 Shijo-cho, Kashihara 634-8522, Nara, Japan; takuya@naramed-u.ac.jp (T.M.); a-shibamoto@naramed-u.ac.jp (A.S.); asahei@naramed-u.ac.jp (S.A.); tomooka@naramed-u.ac.jp (F.T.); kubotaka@naramed-u.ac.jp (T.K.); yuring0309@naramed-u.ac.jp (A.K.); mtanaka@naramed-u.ac.jp (M.T.); satoshi181@naramed-u.ac.jp (S.I.); tsujih@naramed-u.ac.jp (Y.T.); fujinaga@naramed-u.ac.jp (Y.F.); nishimuran@naramed-u.ac.jp (N.N.); shinyasato@naramed-u.ac.jp (S.S.); kitagawa@naramed-u.ac.jp (K.K.); kajik@naramed-u.ac.jp (K.K.); mitoroak@naramed-u.ac.jp (A.M.); htky@naramed-u.ac.jp (H.T.); rnoguchi@naramed-u.ac.jp (R.N.); stakemi@naramed-u.ac.jp (T.A.); yoshijih@naramed-u.ac.jp (H.Y.); 2Department of Evidence-Based Medicine, Nara Medical University, 840 Shijo-cho, Kashihara 634-8522, Nara, Japan; tkinoue0@naramed-u.ac.jp; 3Clinical Research Center, Nara Medical University, Nara Medical University, 840 Shijo-cho, Kashihara 634-8522, Nara, Japan; kasada@naramed-u.ac.jp

**Keywords:** covert hepatic encephalopathy, zinc, neuropsychological test

## Abstract

Covert hepatic encephalopathy (CHE) can worsen the quality of life and prognosis of patients with cirrhosis. We analyzed the risk factors of CHE and identified patients at high risk for overt hepatic encephalopathy (HE) who would benefit from therapeutic interventions. We included 145 patients without a history of or treatment for overt HE. Patients were divided into the CHE and no-CHE groups (*n* = 91 and 54, respectively). CHE had a score above the age-based cutoff value of one of the neuropsychological tests, such as the Stroop and number connection tests. CHE prevalence was 62.8% (*n* = 91). Compared with the no-CHE group, the CHE group had significantly lower serum zinc and albumin levels. Multiple logistic regression analysis identified serum zinc levels at a cutoff value of 74 µg/dL. Subclinical zinc deficiency showed a diagnostic performance of 55.6% sensitivity and 81.5% specificity for CHE. Blood ammonia levels and liver functional reserves were not predictive of CHE. Compared with patients with zinc levels < 74 µg/dL (*n* = 102), those with ≥74 µg/dL (*n* = 43) had significantly lower CHE prevalence and better hepatic functional reserve. Subclinical zinc deficiency was associated with CHE occurrence in patients with cirrhosis without a history of or treatment for overt HE. Measurement of zinc levels facilitates early detection of CHE by neuropsychological testing.

## 1. Introduction

Hepatic encephalopathy (HE) is a neurocognitive disorder resulting from impaired ammonia metabolism in patients with cirrhosis. Covert hepatic encephalopathy (CHE) adversely affects the quality of life (QOL) and prognosis of patients with cirrhosis and frequently leads to overt HE [1]. Despite the absence of obvious clinical symptoms, CHE can subtly impair cognitive function, attention, and motor skills and affect daily activities, such as driving vehicles and work performance. Early detection and management of CHE are crucial for preventing progression to OHE, which is associated with higher risks of hospitalization and mortality. The International Society for Hepatic Encephalopathy and Nitrogen Metabolism consensus guidelines provide a structured approach to CHE diagnosis, emphasizing standardized neurophysiological tests. A key recommendation was CHE screening for patients with cirrhosis who had a previous episode of OHE, developed further decompensation, or have jobs requiring mental alertness.

Psychometric tests are the cornerstone of CHE diagnosis. The psychometric HE score is considered the most used and easily applied approach to minimal HE (MHE) and includes five psychometric tests, such as the number connection tests (NCT) A and B, line tracing test, digit symbol test, and serial dotting test [2]. The Stroop EncephalApp is a quick and simple tool with good sensitivity, reliability, and validity for CHE screening [3]. QuickStroop, which is a newer version of the EncephalApp, includes two runs of the Off State in EncephalApp and can predict CHE/MHE at an equivalent ability with the gold standard within 1 min, compared with the entire time required by the EncephalApp [4]. However, there is no single gold standard diagnostic test or laboratory marker for CHE diagnosis.

Zinc plays a role in synaptic transmission and oxidative stress reduction [5,6], both of which are critical for maintaining cognitive function. Patients with cirrhosis often develop zinc deficiency secondary to reduced intestinal absorption, increased urinary excretion, and portosystemic shunting. Along with ornithine transcarbamylase, which is a key enzyme in the urea cycle, and glutamine synthetase, zinc plays a pivotal role in ammonia detoxification. Zinc deficiency increases blood ammonia levels, leading to cognitive dysfunction associated with HE. Zinc plays a crucial role in blood-brain barrier integrity, particularly relevant in hepatic encephalopathy [7]. Several studies have recently demonstrated that zinc deficiency predicted OHE and mortality in patients with cirrhosis [8,9]. New evidence revealed a strong relationship between serum zinc level and HE severity in patients with decompensated liver cirrhosis [10]. This study examined whether serum zinc level can serve as a diagnostic biomarker for CHE and explored the role of zinc measurement and subsequent neurophysiological tests in detecting CHE in patients with cirrhosis.

## 2. Results

### 2.1. Comparison of Clinical Characteristics Between the CHE and No-CHE Groups

Dietary protein intake was not significantly different among the patients. A total of 145 consecutive patients with cirrhosis were analyzed (Table 1). Compared with the no-CHE group, the CHE group was significantly younger (*p* = 0.042); had significantly lower serum levels of albumin (*p* = 0.033) and zinc (*p* < 0.001) and handgrip strength in women (*p* = 0.030); and significantly higher body mass index (*p* = 0.0045), modified albumin bilirubin score (mALBI) (*p* = 0.032), and serum levels of ammonia (*p* = 0.0025), type IV collagen 7S (*p* = 0.0036), and type III procollagen-N-peptide (*p* = 0.045).

### 2.2. Risk Factors for CHE in Patients with Cirrhosis

Univariate analysis identified that the risk factors for CHE were Child–Pugh score ≥ 8 (odds ratio [OR] 2.22, 95% CI 1.01–4.89, *p* < 0.047); ALBI score ≥ −1.8 (OR 2.98, 95% CI 1.20–7.42, *p* = 0.019); PT < 65% (OR 3.11, 95% CI 1.18–8.18, *p* = 0.022); albumin < 3.1 g/dL (OR 4.73, 95% CI 1.55–14.5, *p* < 0.00); cholinesterase < 179 U/L (OR 2.13, 95% CI 1.04–4.35, *p* = 0.038); branched-chain amino acid to tyrosine ratio (BTR) < 3.8 (OR 2.5, 95% CI 1.17–5.34, *p* = 0.018); zinc < 74 μg/dL (OR 5.10, 95% CI 2.35–11.1, *p* < 0.001); 25-hydroxyvitamin D < 16.5 ng/mL (OR 2.52, 95% CI 1.11–5.69, *p* = 0.027); and type IV collagen 7S ≥ 8.8 ng/mL (OR 3.24, 95% CI 1.52–6.90, *p* = 0.0023). Multivariate analysis identified zinc <74 μg/dL (OR 3.22, 95% CI 1.33–7.76, *p* = 0.0093) as the only predictor of CHE (Table 2).

### 2.3. Diagnostic Accuracy of Zinc Level for CHE

ROC analysis revealed that a cutoff of a zinc level < 74 μg/dL predicted CHE with fair diagnostic performance, with sensitivity of 55.6%, specificity of 81.5%, and area under the ROC curve of 0.705 (95% CI 0.616–0.794) (Figure 1).

### 2.4. Clinical Characteristics According to Zinc Level

Patients were divided into the following two groups based on the cutoff zinc level: high-zinc (≥74 µg/mL, *n* = 43) and low-zinc (<74 µg/mL, *n* = 102) groups. Compared with the low-zinc group, the high-zinc group had significantly lower Child–Pugh scores, ALBI scores, prevalence of CHE, and serum levels of type IV collagen 7S and P-III-P (all *p* < 0.001) and significantly higher PT (*p* < 0.001), cholinesterase (*p* < 0.001), BTR (*p* < 0.001), and serum levels of albumin (*p* < 0.001) and vitamin D (*p* = 0.0015) (Table 3).

## 3. Discussion

In this study, we evaluated the prevalence and risk factors of CHE detected by neuropsychological tests in patients with cirrhosis who had no history of or treatment for OHE. To the best of our knowledge, this was the first study to show an increased risk of CHE in patients with cirrhosis with zinc deficiency levels over 74 µg/dL. The prevalence of CHE was 62.8% overall and was significantly lower in patients with serum zinc levels < 74 µg/dL than in those with serum zinc levels ≥ 74 µg/dL. Previous reports showed that the prevalence of CHE in patients with cirrhosis increased from approximately 20% to 60% as the hepatic functional reserve deteriorated [11,12]. Contrary to the frequency of CHE in our cohort, a recent study reported a 32.7% CHE prevalence in Japanese patients with cirrhosis [13], and a meta-analysis reported a 40% incidence of CHE in patients with cirrhosis [14]. A large multicenter study demonstrated that the incidence of CHE was approximately 35% among 1900 patients with cirrhosis [15]. The discrepancies in CHE incidence may be explained by the different diagnostic tests for CHE and cutoff zinc values used to predict CHE among the studies. Nevertheless, these findings suggested that CHE is the most common complication in patients with cirrhosis.

An albumin level < 3.05 g/dL is associated with the development of CHE in patients with Child–Pugh B cirrhosis [16]. A multicenter study demonstrated that an albumin level < 3.2 g/dL was associated with low performance in the Stroop test among Japanese patients with cirrhosis [17]. However, in this study, no relationship was observed between albumin levels and the presence of CHE. The reason for this difference in the predictive parameters can be explained by the fact that our cohort had an average serum albumin level of 3.76 g/dL, suggesting better hepatic function in our cohort than in previous studies.

Zinc is the second most abundant essential trace element in the human body and plays an indispensable role in various cellular functions and nutrient metabolism [18]. Subclinical zinc deficiency can occur during periods of fluctuating dietary zinc supply and increased demand during inflammation and stress [19]. Patients with advanced-stage liver disease frequently have low serum zinc levels, which are inversely correlated with serum ammonia levels in patients with cirrhosis [20]. Subclinical zinc deficiency can occur as branched chain amino acid (BCAA) consumption in glutamine synthesis increases during progression of chronic liver disease [21]. Subclinical zinc deficiency does not present with visible signs. However, CHE has been associated with falls, motor vehicle accidents, and worsened liver-related outcomes and cirrhosis [14]. A previous study has reported that zinc levels > 80 μg/dL can reduce sarcopenia in patients with cirrhosis [22]. Practice guidelines recommend zinc supplementation even for patients with subclinical zinc deficiency [23,24]. Tomita et al. have proposed a threshold of 80 µg/dL for subclinical zinc deficiency in a Japanese population [25]. These findings supported the hypothesis that maintaining serum zinc levels > 80 μg/dL can prevent CHE in patients with cirrhosis. Further, they indicated that patients with subclinical zinc deficiency may develop medical complications, such as CHE. However, other studies have reported that serum zinc levels < 60 μg/dL and <70 μg/dL were associated with the presence of CHE [13] and a risk factor for CHE in patients with Child–Pugh A cirrhosis [26], respectively. These different thresholds are most likely because this present cohort comprised patients with relatively favorable hepatic function and no history of OHE. Nevertheless, further investigations are needed to investigate the diagnostic performance of a single neuropsychological test for CHE and the impact of subclinical zinc deficiency on QOL, prognosis, and risk for OHE.

Accumulating evidence indicates that disturbances in trace element homeostasis—particularly zinc deficiency—play an essential role in disrupting ammonia metabolism and promoting neuroinflammation. Similarly, increased oxidative stress, which is caused by both systemic inflammation and impaired antioxidant defense mechanisms, has been implicated in the pathophysiology of HE via mechanisms such as astrocyte dysfunction and blood-brain barrier disruption. Shen et al. performed a systematic review and meta-analysis. Results showed that zinc supplementation significantly reduced the incidence of overt HE in patients with cirrhosis [27]. Moreover, experimental models have shown that zinc-deficient mice with hepatic failure exhibit greater neuroinflammation, upregulation of proinflammatory cytokines (e.g., TNF-α, IL-6), and worse performance in spatial memory tests compared with zinc-replete controls [28]. These findings reinforce the hypothesis that zinc deficiency exacerbates cerebral metabolic dysregulation and cognitive decline. In addition, oxidative stress markers such as malondialdehyde and 8-hydroxy-2′-deoxyguanosine are consistently elevated in patients with cirrhosis and are correlated with the severity of neuropsychiatric impairment, thereby supporting the role of reactive oxygen species in the pathogenesis of HE [29,30] Taken together, this underscores the need for a more holistic view of the interplay between trace element deficiencies, oxidative stress, and hepatic encephalopathy, ultimately paving the way for novel therapeutic approaches in managing cirrhosis-related neurocognitive impairment. This integrative perspective supports the rationale for future clinical investigations into combined strategies—such as zinc supplementation, antioxidant therapies, and gut-targeted interventions—to ameliorate neuropsychiatric symptoms and improve patient outcomes. The molecular pathways linking micronutrient imbalance to oxidative stress and neural injury should be further elucidated, as this is essential in refining these therapeutic approaches.

The association between liver fibrosis and CHE has been increasingly recognized. As fibrosis progresses, portosystemic shunting and hepatic functional reserve decline, leading to reduced clearance of neurotoxins such as ammonia. In this context [31], elevated levels of type IV collagen 7S and P-III-P may indirectly indicate a higher risk of neurocognitive impairment, even in the absence of OHE. Further, recent evidence suggests that fibrogenesis is not only a marker of structural liver damage but also a contributor to systemic inflammation, which may exacerbate blood–brain barrier permeability and promote neuroinflammation—both of which are implicated in the development of CHE [32]. Therefore, the observed elevations in type IV collagen 7S and P-III-P in patients with CHE may reflect an underlying fibrotic and inflammatory milieu that contributes to the subclinical neurocognitive dysfunction characteristic of this condition. These findings support the potential utility of fibrosis markers in risk stratification and the early identification of CHE. Clinical studies have shown that zinc supplementation can improve ammonia metabolism and neuropsychological performance, thereby supporting a mechanistic association between zinc status and neuropsychiatric outcomes [32].

In this context, assessing zinc levels in patients with liver cirrhosis may provide valuable insights into CHE risk and guide early nutritional interventions. Our findings reinforce the growing recognition of zinc as a modifiable factor in the pathophysiology of CHE and emphasize the need for further clinical research to evaluate its diagnostic and therapeutic utility.

This study had several limitations. First, this was a retrospective study conducted at a single institution with a restricted number of patients. The study focused on patients with no history of OHE and had good hepatic functional reserve. Second, there is no gold standard test for CHE diagnosis, although various combinations of psychometric and neuropsychological tests have been proposed. The EASL and AASLD guidelines recommend the animal naming test for CHE detection [33]. However, further investigation is required to determine the optimal test for the diagnosis of CHE [34]. Nevertheless, this study showed that 62.8% of patients with cirrhosis who had no prior history of or treatment for OHE developed CHE and that a serum zinc level < 74 µg/dL was a predictor of CHE. Zinc supplementation may improve the QOL of patients with cirrhosis who complain of falls and driving difficulties and encounter traffic accidents and may prolong survival by correcting zinc deficiency, which reduces urea cycle function.

In conclusion, subclinical zinc deficiency was associated with the occurrence of CHE in patients with cirrhosis who had no history of or therapy for OHE. Measurement of zinc levels facilitated early detection of CHE by neuropsychological testing.

## 4. Patients and Methods

A single-center cohort study on outpatients with cirrhosis was performed between January 2011 and March 2023 at Nara Medical University Hospital. Of 823 patients, 678 were excluded because of age > 80 years (*n* = 217), presence or history of OHE (*n* = 95), presence of portosystemic shunt > 5 mm (*n* = 5), neurological or psychiatric comorbidities (*n* = 35), uncontrolled infection (*n* = 22), uncontrolled hepatocellular carcinoma (HCC) (*n* = 26), extrahepatic malignancies (*n* = 21), liver transplantation (*n* = 2), a lack of data on laboratory tests (*n* = 208), history of alcohol consumption (>60 g per day in men and 40 g per day in women) from 3 months up to the present (*n* = 10), and prescription of probiotics (*n* = 37) (Figure 2). None of the patients in this study were treated with lactulose, rifaximin, metronidazole, or other drugs (including benzodiazepines, antiepileptics, and psychotropics) that affect psychometric performances.

Liver cirrhosis was diagnosed according to clinical data, including laboratory tests, medical imaging features, liver histology, and clinical complications, such as HE and ascites. Hand grip strength (HGS) was measured on admission, and clinical parameters and serum assay for endotoxin activity were evaluated in 151 patients with cirrhosis. All enrolled patients received dietary advice from dietitians. Zinc levels were measured using a colorimetric assay kit (Abcam, ab102507 Melbourne, Australia), according to the manufacturer’s instructions. Absorbance was measured at 560 nm [35].

To objectively evaluate cognitive performance, we used a neuropsychological test, such as the Stroop test or NCT-A and NCT-B by the Japan Society of Hepatology, as previously described [36]. The hardware for neuropsychological tests comprised a touch screen tablet, such as an iPad (Apple, Cupertino, CA, USA). The Stroop test (EncephalApp) was administered using the EncephalApp (available on iOS/Android platforms), which consists of the following two components: the “off” state (control) and the “on” state (interference). In the off state, the participants were instructed to identify the color of the solid-colored bars as immediately as possible. In the on state, the participants were shown color words (e.g., “red” and “blue”) printed in incongruent ink colors and were instructed to identify the font color while disregarding the word itself. Each component included 5 runs of 10 stimuli each, preceded by a training session to ensure task comprehension. The total time to complete the off-and-on tasks, as well as the number of errors, were recorded automatically by the application. The on-time score was used as the primary measure of cognitive interference. The established cutoff values adjusted for age and education were used to identify cognitive impairment indicative of CHE. The Number Connection Test (NCT-A/B) was administered using standardized paper-based forms. In the NCT-A, patients were instructed to connect 25 randomly arranged numbers in ascending order (from 1 to 25) as quickly as possible. In the NCT-B, which imposes a higher cognitive load, patients were instructed to alternate between numbers and letters in ascending order up to the endpoint. The age-specific cutoff values of the respective neuropsychological tests are currently available on the website of the Japan Society of Hepatology.

The time taken to complete each task was measured in seconds, and any errors in sequence were documented. Normative data adjusted for age and education level were used to interpret the results. Prolonged completion time or sequencing errors were considered indicative of cognitive dysfunction. Patients were diagnosed with CHE when one of the neuropsychological tests was positive. The patients were divided into two group: those who developed CHE (CHE group) and those who did not develop CHE (no-CHE group).

### Statistical Analyses

All statistical analyses were conducted using EZR (Saitama Medical Center, Jichi Medical University), version 1.68 [37]. Normally and non-normally distributed continuous variables were expressed as mean and standard deviation or as median and interquartile range (25–75%), respectively. Baseline characteristics were compared between the CHE and no-CHE groups using an unpaired *t*-test or Mann–Whitney *U* test. Parametric tests were used for normally distributed continuous data, whereas nonparametric tests were used for non-normally distributed data. A two-sided *p* < 0.05 was considered statistically significant. Receiver operating characteristic (ROC) curves in logistic regression were used to determine the optimal cutoff values of the continuous variable parameters and to identify the predictive threshold value of serum zinc for CHE.

## Figures and Tables

**Figure 1 ijms-26-04184-f001:**
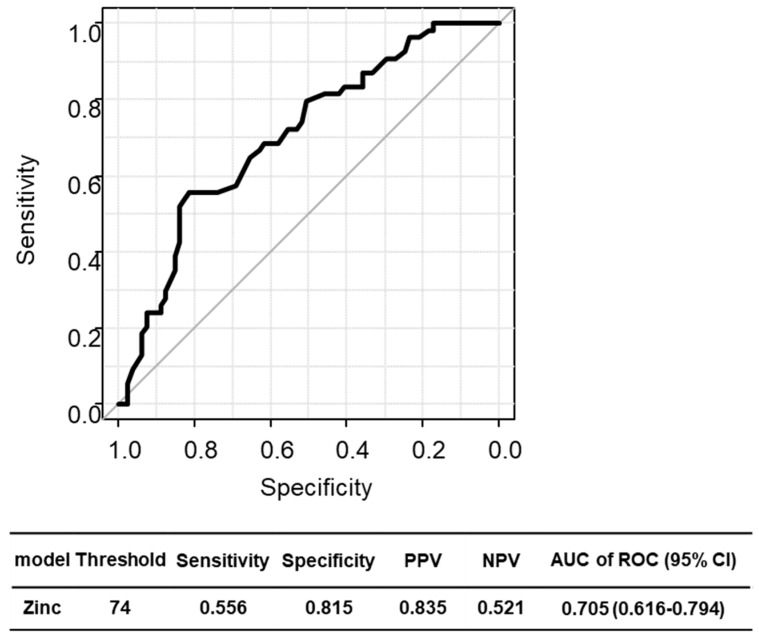
ROC curve for the diagnostic accuracy of zinc level for CHE in patients with cirrhosis. CHE, covert hepatic encephalopathy; OHE, overt hepatic encephalopathy; HCC, hepatocellular carcinoma; ROC, receiver operating characteristic.

**Figure 2 ijms-26-04184-f002:**
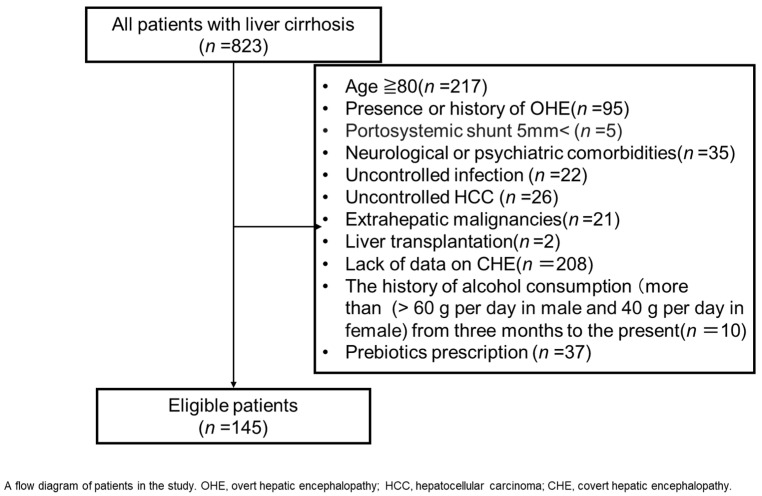
Study scheme.

**Table 1 ijms-26-04184-t001:** Clinical characteristics of patients with cirrhosis according to CHE and no CHE.

Variables	All Patients (*n* = 145)	No-CHE (*n* = 54)	CHE (*n* = 91)	*p* ^a^
Age, years	66.7 ± 11.5	69.2 ± 8.8	65.2 ± 12.6	0.052
Male, *n* (%)	86 (59.3)	30 (55.6)	56 (61.5)	0.49
BMI, kg/m^2^	25.0 ± 4.7	23.6 ± 4.0	25.9 ± 4.9	0.0045
Etiology HCV/HBV/ALD/NASH/other ^b^	32/18/32/29/34	8/8/12/11/15	24/10/20/18/19	0.53
HCC, *n* (%)	18 (12.4)	7 (13.0)	11 (12.1)	1
Esophagogastric varices, *n* (%)	83 (57.2)	33 (61.1)	50 (54.9)	0.49
Ascites, *n* (%)	38 (26.2)	13 (24.1)	25 (27.5)	0.70
Child-Pugh score	6 (5–8)	6 (5–7)	6 (5–9)	0.051
Child-Pugh class (A/B/C)	86/42/17	36/16/2	50/26/15	0.060
mALBI score	−2.27 ± 0.75	−2.45 ± 0.50	−2.17 ± 0.85	0.032
HGS, kg				
Male	31.8 ± 10.1	30.8 ± 9.7	32.4 ± 10.3	0.52
Female	17.5 ± 6.1	19.5 ± 5.6	15.9 ± 6.0	0.030
Reduced HGS, %	41.1	36.7	44.0	0.46
PT, %	83.9 ± 22.5	88.4 ± 19.3	81.2 ± 23.9	0.065
Platelet, ×10^4^/μL	11.6 ± 6.0	11.7 ± 6.1	11.5 ± 5.9	0.89
BUN, mg/dL	18.2 ± 11.0	16.7 ± 9.0	19.0 ± 11.9	0.21
Creatinine, mg/dL	0.80 (0.65–0.98)	0.76 (0.64–0.91)	0.83 (0.66–1.06)	0.17
AST, U/L	35 (26–45)	36 (25–42)	35 (26–49)	0.40
ALT, U/L	23 (17–31)	23 (16–30)	23 (18–35)	0.17
γ-GTP, mg/dL	49 (23–98)	50 (21–117)	48 (24–87)	0.79
ALP, U/L	117 (85–230)	112 (86–195)	117 (85–252)	0.61
Albumin, g/dL	3.76 ± 0.73	3.93 ± 0.54	3.67 ± 0.80	0.033
Total Bilirubin, mg/dL	1.3 (1.0–2.1)	1.2 (1.0–1.9)	1.4 (1.0–2.3)	0.43
ChE, U/L	209 ± 97	219 ± 90	204 ± 100	0.34
NH3, μg/dL	52.5 ± 42.5	38.6 ± 21.5	60.7 ± 49.3	0.0025
BTR	4.76 ± 2.00	5.12 ± 1.95	4.50 ± 2.01	0.079
Zinc, μg/dL	65.6 ± 18.5	73.0 ± 13.9	60.7 ± 19.7	<0.001
EAA	0.292 ± 0.121	0.305 ± 0.117	0.285 ± 0.124	0.37
AFP, ng/mL	3.5 (2.3–5.9)	3.9 (2.5–6.2)	3.4 (2.2–5.1)	0.33
25-hydroxyvitamin D, ng/mL	15.1 ± 6.3	15.7 ± 6.1	14.7 ± 6.4	0.39
Type IV collagen 7S, ng/mL	8.88 ± 7.54	6.55 ± 2.89	10.4 ± 9.12	0.0036
Type III procollagen-N-peptide, U/m	0.914 ± 0.543	0.823 ± 0.329	1.01 ± 0.64	0.045
M2BPGi, C.O.I	2.55 (1.12–4.84)	2.41 (1.06–4.54)	2.80 (1.23–5.45)	0.29
FIB-4 index	4.87 (2.83–7.14)	4.82 (3.33–6.62)	4.91 (2.74–7.29)	0.99
APRI	1.10 (0.61–1.96)	1.00 (0.61–1.59)	1.18 (0.61–2.03)	0.39

Categorical data are presented as numbers. Continuous data are presented as the mean (±standard deviation) or median (interquartile range). ^a^ Differences between the two groups were analyzed using the χ^2^-test or Student’s *t*-test. ^b^ Includes autoimmune hepatitis and primary biliary cholangitis. BMI, body mass index; HCV, hepatitis C virus; HBV, hepatitis B virus; ALD, alcohol-related liver disease; NASH, nonalcoholic steatohepatitis; HCC, hepatocellular carcinoma; m-ALBI, modified albumin-bilirubin; HGS, handgrip strength; PT, prothrombin time; BUN, blood urea nitrogen; AST, aspartate aminotransferase; ALT, alanine aminotransferase; γ-GTP, γ-glutamyl transpeptidase; ALP, alkaline phosphatase; ChE, cholinesterase; NH3, ammonia; BTR, branched-chain amino acid to tyrosine ratio; EAA, endotoxin activity assay; AFP, α-fetoprotein; M2BPGi, mac-2 binding protein glycosylation isomer; FIB-4, fibrosis 4; APRI, aspartate aminotransferase to platelet ratio index.

**Table 2 ijms-26-04184-t002:** Risk factors for covert hepatic encephalopathy in patients with cirrhosis.

	Univariate Analysis		Multivariate Analysis	
	OR (95% CI)	*p*	OR (95% CI)	*p*
Age ≤ 63, Years	2.09 (0.97–4.51)	0.061		
BMI ≥ 25.5, kg/m^2^	1.86 (0.91–3.81)	0.089		
Child-Pugh score ≥ 8	2.22 (1.01–4.89)	0.047		
mALBI score ≥ −1.8	2.98 (1.20–7.42)	0.019		
HGS, kg				
Male ≤ 28	0.70 (0.25–1.94)	0.49		
Female ≤ 18	2.83 (0.93–8.62)	0.068		
PT < 65, %	3.11 (1.18–8.18)	0.022	1.56 (0.43–5.62)	0.50
Platelet < 4.1, ×10^4^/μL	3.74 (0.44–31.9)	0.23		
BUN ≥ 14.0, mg/dL	1.26 (0.62–2.55)	0.53		
Creatinine ≥ 1.04, mg/dL	2.27 (0.90–5.72)	0.082		
AST ≥ 37, U/L	0.74 (0.32–1.71)	0.48		
ALT ≥ 25, U/L	1.18 (0.59–2.34)	0.64		
γ-GTP ≥ 74, mg/dL	0.50 (0.25–1.01)	0.054		
ALP ≥ 92, U/L	1.55 (0.740–3.23)	0.25		
Albumin < 3.1, g/dL	4.73 (1.55–14.5)	<0.001	2.16 (0.49–9.55)	0.31
Total bilirubin ≥ 2.2, mg/dL	1.89 (0.81–4.44)	0.141		
ChE < 179, U/L	2.13 (1.04–4.35)	0.038		
NH3 ≥ 77.1, μg/dL	4.52 (1.47–13.9)	0.0084	1.62 (0.45–5.86)	0.46
BTR < 3.8	2.5 (1.17–5.34)	0.018		
Zinc < 74, μg/dL	5.10 (2.35–11.1)	<0.001	3.22 (1.33–7.76)	0.0093
EAA ≥ 0.21	0.56 (0.23–1.32)	0.18		
AFP ≥ 3.8, ng/mL	0.51 (0.25–1.04)	0.065		
25-hydroxyvitamin D < 16.5, ng/mL	2.52 (1.11–5.69)	0.027		
Type IV collagen 7S ≥ 8.8, ng/mL	3.24 (1.52–6.90)	0.0023	1.59 (0.59–4.30)	0.36
P-III-P ≥ 1.0, U/m	1.77 (0.80–3.92)	0.157		
M2BPGi ≥ 6.5, C.O.I	3.65 (0.98 –13.7)	0.054		
FIB4 index ≥ 6.1	1.38 (0.67–2.84)	0.38		
APRI ≥ 1.7	1.86 (0.85–4.07)	0.12		

BMI, body mass index; CI, confidence interval; OR, odds ratio; m-ALBI, modified Albumin-bilirubin; HGS, handgrip strength; PT, prothrombin time; BUN, blood urea nitrogen; AST, aspartate aminotransferase; ALT, alanine aminotransferase; γ-GTP, γ-glutamyl transpeptidase; ALP, alkaline phosphatase; ChE, cholinesterase; NH3, ammonia; BTR, branched-chain amino acid to tyrosine ratio; EAA, Endotoxin Activity Assay; AFP, α-fetoprotein; P-III-P, type III procollagen-N-peptide; M2BPGi, mac-2 binding protein glycosylation isomer; FIB-4, fibrosis 4; APRI, aspartate aminotransferase to platelet ratio index.

**Table 3 ijms-26-04184-t003:** Clinical characteristics of patients with cirrhosis according to zinc levels 74 μg/dL.

Variables	All Patients (*n* = 145)	Zn ≥ 74 (*n* = 43)	Zn < 74 (*n* = 102)	*p* ^a^
Age, years	66.7 ± 11.5	68.8 ± 8.8	65.8 ± 12.4	0.15
Male, *n* (%)	86 (59.3)	23 (53.4)	63 (61.8)	0.36
BMI, kg/m^2^	25.0 ± 4.7	23.9 ± 4.0	25.5 ± 4.9	0.073
Etiology HCV/HBV/ALD/NASH/other ^b^	32/18/32/29/34	12/10/6/7/8	20/8/26/22/26	0.058
HCC, *n* (%)	18 (12.4)	4 (9.3)	14 (13.7)	0.59
Esophagogastric varices, *n* (%)	83 (57.2)	24 (55.8)	59 (57.8)	0.71
Ascites, *n* (%)	38 (26.2)	7 (16.3)	31 (30.4)	0.099
Child-Pugh score	6 (5–8)	6 (5–7)	6 (5–9)	0.051
Child-Pugh class (A/B/C)	86/42/17	34/9/0	52/33/17	<0.001
mALBI score	−2.27 ± 0.75	−2.74 ± 0.43	−2.08 ± 0.77	<0.001
HGS, kg				
Male	31.8 ± 10.1	33.4 ± 9.2	31.1 ± 10.5	0.38
Female	17.5 ± 6.1	18.7 ± 6.2	16.8 ± 6.0	0.28
Reduced HGS, %	41.1	55.4	65.9	0.33
CHE, *n* (%)	91 (62.8)	14 (32.6)	77 (75.5)	<0.001
PT, %	83.9 ± 22.5	94.2 ± 22.5	79.5 ± 22.9	<0.001
Platelet, ×10^4^/μL	11.6 ± 6.0	12.9 ± 6.0	11.0 ± 5.9	0.078
BUN, mg/dL	18.2 ± 11.0	17.3 ± 5.3	18.5 ± 12.6	0.54
Creatinine, mg/dL	0.80 (0.65–0.98)	0.76 (0.64–0.91)	0.83 (0.66–1.06)	0.17
AST, U/L	35 (26–45)	36 (25–42)	35 (26–49)	0.40
ALT, U/L	23 (17–31)	23 (16–30)	23 (18–35)	0.17
γ-GTP, mg/dL	49 (23–98)	50 (21–117)	48 (24–87)	0.79
ALP, U/L	117 (85–230)	112 (86–195)	117 (85–252)	0.61
Albumin, g/dL	3.76 ± 0.73	4.23 ± 0.48	3.57 ± 0.72	<0.001
Total bilirubin, mg/dL	1.3 (1.0–2.1)	1.2 (1.0–1.9)	1.4 (1.0–2.3)	0.43
ChE, U/L	209 ± 97	259 ± 88	189 ± 93	<0.001
NH3, μg/dL	52.5 ± 42.5	33.9 ± 23.1	60.3 ± 46.2	<0.001
BTR	4.76 ± 2.00	5.68 ± 1.71	4.32 ± 1.99	<0.001
EAA	0.292 ± 0.121	0.271 ± 0.116	0.302 ± 0.123	0.19
AFP, ng/mL	3.5 (2.3–5.9)	3.9 (2.5–6.2)	3.4 (2.2–5.1)	0.33
25-hydroxyvitamin D, ng/mL	15.1 ± 6.3	17.8 ± 7.5	13.8 ± 5.1	0.0015
Type IV collagen 7S, ng/mL	8.88 ± 7.54	5.46 ± 2.34	10.50 ± 8.55	<0.001
P-III-P, U/m	0.914 ± 0.543	0.650 ± 0.203	1.037 ± 0.606	<0.001
M2BPGi, C.O.I	2.55 (1.12–4.84)	2.41 (1.06–4.54)	2.80 (1.23–5.45)	0.29
FIB4 index	4.87 (2.83–7.14)	4.82 (3.33–6.62)	4.91 (2.74–7.29)	1.0
APRI	1.10 (0.61–1.96)	1.00 (0.61–1.59)	1.18 (0.61–2.03)	0.39

Categorical data are presented as numbers. Continuous data are presented as the mean (±standard deviation) or median (interquartile range). ^a^ Differences between the two groups were analyzed using the χ^2^-test or Student’s *t*-test. ^b^ Includes autoimmune hepatitis and primary biliary cholangitis. BMI, body mass index; HCV, hepatitis C virus; HBV, hepatitis B virus; ALD, alcohol-related liver disease; NASH, nonalcoholic steatohepatitis; HCC, hepatocellular carcinoma; m-ALBI, modified albumin-bilirubin; HGS, handgrip strength; CHE, covert hepatic encephalopathy; PT, prothrombin time; BUN, blood urea nitrogen; AST, aspartate aminotransferase; ALT, alanine aminotransferase; γ-GTP, γ-glutamyl transpeptidase; ALP, alkaline phosphatase; ChE, cholinesterase; NH3, ammonia; BTR, branched-chain amino acid to tyrosine ratio; EAA, endotoxin activity assay; AFP, α-fetoprotein; P-III-P, type III procollagen-N-peptide; M2BPGi, mac-2 binding protein glycosylation isomer; FIB-4, fibrosis 4; APRI, aspartate aminotransferase to platelet ratio index.

## Data Availability

Raw data were generated at Nara University Hospital. The datasets generated and/or analyzed during the current study are available from the corresponding author (T.N.) upon reasonable request.

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
