# Peer review of "Clinical Significance of Marginal Zinc Deficiency as a Predictor of Covert Hepatic Encephalopathy in Patients with Liver Cirrhosis"

_ijms, 2025, doi:10.3390/ijms26094184_

Round 1

Reviewer 1 Report

Comments and Suggestions for Authors

The manuscript under review provides a thorough investigation into the clinical significance of marginal zinc deficiency as a predictor of covert hepatic encephalopathy (CHE) in patients with liver cirrhosis. The study’s design is methodologically sound, encompassing a well‐characterized cohort of cirrhotic patients without a prior history of overt hepatic encephalopathy (OHE), and employs a combination of clinical assessments, biochemical measurements, and neuropsychological testing to evaluate the association between serum zinc levels and the occurrence of CHE. The authors have demonstrated that a serum zinc level below 74 μg/dL is an independent predictor of CHE, as shown by multivariate logistic regression analysis, and have reinforced the clinical relevance of zinc in the context of hepatic encephalopathy through robust statistical analyses, including ROC curve evaluation and subgroup comparisons.

One of the strengths of this manuscript is the comprehensive presentation of clinical data; the authors provide detailed tables comparing various parameters between CHE and non-CHE groups, thereby allowing for a clear understanding of the differences in liver function, nutritional status, and neuromuscular performance between these patient subsets. The use of standardized neuropsychological tests for CHE diagnosis further enhances the study’s rigor by offering an objective measure of cognitive function in a population that may otherwise display subtle neurocognitive impairments.

Moreover, the discussion adequately contextualizes the findings within the broader landscape of cirrhosis complications by linking zinc deficiency to impaired ammonia detoxification and subsequent neurocognitive dysfunction. The authors explore potential mechanisms by which subclinical zinc deficiency might contribute to the pathogenesis of CHE, and they highlight the importance of early zinc supplementation as a potential interventional strategy to improve the quality of life and clinical outcomes in these patients.

However, while the manuscript presents valuable insights into the role of zinc in CHE, the discussion could be further enriched by incorporating additional literature that elucidates the underlying molecular mechanisms, particularly those related to oxidative stress and the antioxidant defense system. In this regard, the inclusion of studies such as the one with DOI: 10.3390/antiox13050609 and the article with DOI: 10.3390/antiox13070841 would be highly beneficial. These references address the relationship between oxidative stress, zinc status, and cellular antioxidant mechanisms, thereby providing a more comprehensive backdrop for understanding how marginal zinc deficiency might exacerbate neurocognitive dysfunction through increased oxidative damage in cirrhotic patients.

Furthermore, the authors might consider discussing how oxidative stress markers and antioxidant status could potentially serve as complementary predictors or therapeutic targets in patients with cirrhosis. Expanding on this point with data and interpretations from the suggested articles would not only broaden the conceptual framework of the study but also might help to clarify whether zinc supplementation could mitigate oxidative injury and, consequently, improve neural function in this patient population.

In summary, the manuscript is a well-executed study that significantly contributes to our understanding of the clinical implications of marginal zinc deficiency in liver cirrhosis, particularly regarding the development of CHE. The detailed statistical analyses and comprehensive clinical comparisons underscore the robustness of the findings. Nonetheless, the discussion would benefit from an expanded review of the oxidative stress pathways involved, and the inclusion of recent studies—specifically, those referenced by DOI: 10.3390/antiox13050609 and DOI: 10.3390/antiox13070841—is recommended to augment the theoretical context of the research. Such integration would offer a more holistic view of the interplay between trace element deficiencies, oxidative stress, and hepatic encephalopathy, ultimately paving the way for novel therapeutic approaches in the management of cirrhosis-related neurocognitive impairment.

Author Response

Response: I cited the two abovementioned papers (reference numbers 5 and 6) and added some details to the Discussion section. We have included a description of these findings on page 7, lines 186–210.

Reviewer 2 Report

Comments and Suggestions for Authors
  1. There were important points in exclusion criteria. The authors dose not declared if they excluded such as recent history of alcohol intake, lactulose, probiotics, metronidazole, rifaximin, patients on drugs affecting psychometric performances like benzodiazepines, antiepileptics or psychotropic drugs.
  2. The authors must show how they do Stroop test or  NCT in details
  3. In table 3 what is the difference between Zn ≧74 and 74>Zn; both above 74?
  4. Could the authors provide the explanation for the clinical significance of included type IV collagen 7S and P-III P and its relation to CHE?
  5. The authors mentioned; To the best of our knowledge, this was the first study to show an increased risk of CHE in patients with cirrhosis who had subclinical zinc deficiency? Many studies declared before this knowledge and cirrhosis is associated with malnutrition included Zinc deficiency, ever many studies handling treatment of CHE by zinc
  6. The discussion is weak and need to be focused on the role of zinc in CHE

Author Response

Reviewer2

  1. There were important points in exclusion criteria. The authors dose not declared if they excluded such as recent history of alcohol intake, lactulose, probiotics, metronidazole, rifaximin, patients on drugs affecting psychometric performances like benzodiazepines, antiepileptics or psychotropic drugs.

Response: The health history, test results, and other essential details of the patients were re-examined. Further, we found that 255 patients who had missing data, 10 patients who had a history of alcohol consumption (> 60 g per day in men and 40 g per day in women) from 3 months up to the present, and 37 patients who were treated with probiotics were enrolled in the current study. None of the patients in this study received lactulose, rifaximin, metronidazole, or other drugs (including benzodiazepines, antiepileptics, and psychotropics), which affect psychometric performances. Figure 2 was revised.

  1. The authors must show how they do Stroop test or NCT in details

I revised the manuscript.

Response: Some details were added. A description of these findings was included on pages 8–9, lines 268–285.

  1. In table 3 what is the difference between Zn ≧74 and 74>Zn; both above 74?

Response: Table 3 was revised accordingly.

  1. Could the authors provide the explanation for the clinical significance of included type IV collagen 7S and P-III P and its relation to CHE?

The association between liver fibrosis and CHE has been increasingly recognized. With the progression of fibrosis, portosystemic shunting and hepatic functional reserve decline, leading to reduced clearance of neurotoxins such as ammonia. In this context, elevated levels of type IV collagen 7S and P-III P may indirectly indicate a higher risk of neurocognitive impairment, even in the absence of OHE. Further, recent evidence suggests that fibrogenesis is not only a marker of structural liver damage but also a contributor to systemic inflammation, which may exacerbate blood–brain barrier permeability and promote neuroinflammation—both of which are implicated in the development of CHE. Therefore, the observed elevations in type IV collagen 7S and P-III P in patients with CHE may reflect an underlying fibrotic and inflammatory milieu that contributes to the subclinical neurocognitive dysfunction characteristic of this condition. These findings support the possible application of fibrosis markers in risk stratification and the early identification of CHE. Clinical studies have shown that zinc supplementation can improve ammonia metabolism and neuropsychological performance, thereby supporting a mechanistic association between zinc status and neuropsychiatric outcomes. We have included a description of these findings on page 8, lines 211–226.

  1. The authors mentioned; To the best of our knowledge, this was the first study to show an increased risk of CHE in patients with cirrhosis who had subclinical zinc deficiency? Many studies declared before this knowledge and cirrhosis is associated with malnutrition included Zinc deficiency, ever many studies handling treatment of CHE by zinc

Response: The manuscript was revised accordingly. This study first showed that patients with cirrhosis who have zinc deficiency levels > 60 µg/dL were at high risk of CHE. A description of these findings was included on page 7, lines 143–144.

  1. The discussion is weak and need to be focused on the role of zinc in CHE

Clinical studies have shown that zinc supplementation can improve ammonia metabolism and neuropsychological performance. In addition, experimental data have revealed that zinc-deficient mice with hepatic failure exhibit worsened neuroinflammatory responses and cognitive performance. This finding supports a mechanistic association between zinc status and neuropsychiatric outcomes (2).

In this context, assessing zinc levels in patients with liver cirrhosis may provide valuable insights into the risk of CHE and guide early nutritional interventions. Our findings reinforce the growing recognition of zinc as a modifiable factor in the pathophysiology of CHE and emphasize the need for further clinical research to evaluate its diagnostic and therapeutic utility. A description of these findings was included on page 8, lines 223–231.

Round 2

Reviewer 2 Report

Comments and Suggestions for Authors

none

Author Response

n/a
